# Characterization of Synthetic Polymer Coated with Biopolymer Layer with Natural Orange Peel Extract Aimed for Food Packaging

**DOI:** 10.3390/polym15112569

**Published:** 2023-06-02

**Authors:** Domagoj Gabrić, Mia Kurek, Mario Ščetar, Mladen Brnčić, Kata Galić

**Affiliations:** Faculty of Food Technology and Biotechnology, University of Zagreb, Pierottijeva 6, HR-10000 Zagreb, Croatia; dgabric@pbf.hr (D.G.); mscetar@pbf.hr (M.Š.);

**Keywords:** chitosan biolayer, orange peel essential oil, gas and water vapor barrier, overall migration

## Abstract

This research was aimed to make biolayer coatings enriched with orange peel essential oil (OPEO) on synthetic laminate, oriented poly(ethylene-terephthalate)/polypropylene (PET-O/PP). Coating materials were taken from biobased and renewable waste sources, and the developed formulation was targeted for food packaging. The developed materials were characterized for their barrier (O_2_, CO_2_, and water vapour), optical (colour, opacity), surface (inventory of peaks by FTIR), and antimicrobial activity. Furthermore, the overall migration from a base layer (PET-O/PP) in an acetic acid (3% HAc) and ethanol aqueous solution (20% EtOH) were measured. The antimicrobial activity of chitosan (Chi)-coated films was assessed against *Escherichia coli*. Permeation of the uncoated samples (base layer, PET-O/PP) increased with the temperature increase (from 20 °C to 40 °C and 60 °C). Films with Chi-coatings were a better barrier to gases than the control (PET-O/PP) measured at 20 °C. The addition of 1% (*w*/*v*) OPEO to the Chi-coating layer showed a permeance decrease of 67% for CO_2_ and 48% for O_2_. The overall migrations from PET-O/PP in 3% HAc and 20% EtOH were 1.8 and 2.3 mg/dm^2^, respectively. Analysis of spectral bands did not indicate any surface structural changes after exposure to food simulants. Water vapour transmission rate values were increased for Chi-coated samples compared to the control. The total colour difference showed a slight colour change for all coated samples (ΔE > 2). No significant changes in light transmission at 600 nm for samples containing 1% and 2% OLEO were observed. The addition of 4% (*w*/*v*) OPEO was not enough to obtain a bacteriostatic effect, so future research is needed.

## 1. Introduction

Conventional food packaging materials are produced from non-degradable, petrochemical-based materials that lead to substantial environmental pollution and numerous ecological problems, especially if there is no possibility of recycling them. Petroleum-based polymers are mostly used for production of multilayer materials fulfilling barrier and mechanical requirements for food packaging. Such a tailor-made plastic offers better protection and longer shelf life for specific food products. Polypropylene (PP) is one of the most common materials used in the food industry owing to its affordable production cost, good thermal stability, and attractivity to consumers due to its transparency and the visibility of the packaged product [1]. Further, one of the biggest advantages is the adaptability of PP for a wide range of food products [2]. Combing PP with oriented polyethylene terephthalate (PET-O) makes materials with an excellent water vapour barrier and high tear and puncture resistance. In addition, PET-O/PP has good sealability, so it is often used as a top cover for PP buckets.

Permeation is one of the crucial parameters in food preservation and is defined as a molecular movement of gases, vapours, or fluids, which can be described as a combination of three coefficients: permeability (P), solubility (S), and diffusion (D) [3,4]. It is important to emphasise that optimal barrier properties are required for not only synthetic packaging materials, but also biomaterials aimed for specific food products [5,6]. Nowadays, there is an increasing trend to replace plastics with natural biodegradable polymers from renewable agriculture byproducts and food industry wastes [7,8,9]. However, due to their versatility and sensitivity to humid environments, this is often a hardly achievable task. Biodegradable polymers are produced using different treatments, such as chemical processing and treatment by microorganisms and enzymes, in order to make it easier for environmentally friendly disposal after their end usage. An excellent review paper on biodegradable polymers, treatment, composites, blending, and modelling is written by [10]. Multilayer materials with at least one layer of biopolymer or biodegradable composites are the target of many of today’s studies related to issues about the separation, composting, and recycling of multilayer flexible packaging related to [11]. Single-layer materials for food packaging often do not provide a sufficient barrier to all gases, so they must be improved by adding functional barriers. Those multilayer approaches create end-of-life challenges since a complex structure and blending makes those materials impossible to recycle or difficult to biodegrade [12]. The research community, thus, searches for improved barrier layers with adequate biodegradability and/or recyclability to follow the sustainability concept. In the present study, even though the used synthetic laminate would be difficult to recycle as such, it served as a reference material with regards to its barrier characteristics. It also served to explore whether its functionality could be improved with an active compound addition. By incorporating an active compound (antioxidant, antimicrobial, etc.) in the biodegradable polymer, adequate food safety and quality with extended shelf life can be ensured [13,14,15]. The antimicrobial activity of chitosan films was observed on the growth of a wide variety of bacteria (*Bacillus cereus*, *Escherichia coli*, *Salmonella typhimurium*, *Staphylococcus aureus*, *Listeria monocytogenes*, etc.) and fungi (*Botrytis cinerea*, *Fusarium oxysporum*, *Trichophyton equinum*, *Piricularia oryzae*, etc.) [16,17]. Phenols in citrus peels are known to have antimicrobial activity due to their high amounts of polyphenols and were found to be effective against *Staphylococcus aureus*, *Escherichia coli*, *Pseudomonas aeruginosa*, and *Klebsiella pneumoniae* [18], as well as against food-borne pathogens [19].

The present research was carried out in two stages. Primarily, the physico-chemical (colour, transparency) and barrier properties (gas and condensable gas—water vapour) of bi-layered PET-O/PP (referent commercial sample) was determined. Then, referent films were coated with functional biopolymer coatings (chitosan-based) enriched with orange peel essential oil (OPEO) to make antimicrobial films. Gas (O_2_ and CO_2_) permeance was measured at 3 temperatures: 20 °C as a typical storage temperature, and 40 °C and 60 °C as elevated and inadequate storage temperatures simulating conditions during the summer period and/or temperature fluctuation during filling and storage. The overall migration in 2 food simulants (3% (*w*/*v*) acetic acid and 20% (*v*/*v*) ethanol) was determined for referent PET-O/PP samples. Orange peel essential oil was used as a natural preservative, and the antimicrobial activity of a chitosan-enriched biolayer coated on PET-O/PP was tested against *Escherichia coli* (ATTC 25922).

## 2. Materials and Methods

### 2.1. Materials

Commercially available oriented poly(ethylene-terephthalate)/polypropylene (PET-O/PP) laminate (total thickness 62 µm (PET-O/PP: 12/50 µm), transparent and bluish taint) (Coveris, Warburg, Germany) was used as a base layer. Chitosan (type 652, molecular weight 165 kDa, degree of deacetylation above 85%, France Chitin, Marseille, France), glycerine (minimum purity 99.5%, E422, Dekorativna točka d.o.o., Poznanovec, Croatia), and Tween 80 (Sigma-Aldrich Co, St. Louis, MO, USA) were used for the preparation of biopolymer coatings. *Escherichia coli* ATTC 25922 was used for antimicrobial analysis. Oranges (Soursos S.A., Argos, Greece) were purchased on the local market and peeled, and the peels were used for the preparation of orange peel essential oil (OPEO). OPEO was prepared using microwave-assisted extraction operating in Ethos X mode (Ethos Easy Milestone Srl, Sorisole, Italy) [20]. Extracted oil was stored in closed glass vials in the dark and cold (4 °C) before use.

### 2.2. Methods

#### 2.2.1. Preparation of Coated Films

Biopolymer solutions were prepared from chitosan (at 2% *w*/*v*, in 1% aqueous acetic acid (*v*/*v*)), with glycerol as plasticizer (30% of polymer dry weight (w/pdw)) and Tween 80 as emulsifier. For active coatings, OPEO was added in a final step in 3 concentrations: 1, 2, and 4% (*w*/*v*). For homogenization, a handheld ultrasonic homogenizer (UP200Ht, Hielscher Ultrasonics GmbH, Teltow, Germany), operating at 200 W and 26 kHz for 15 min, was used. During the homogenization, samples were cooled in an ice water bath. The final coating compositions are given in Table 1. Freshly prepared coatings were coated on the PP surface of PET-O/PP films using a manual laboratory glass rod. Since the surface properties of polyolefins are not favourable for the coating process, a corona discharge pre-treatment was used before the application of biopolymer coatings (Handheld “Electro-technic products Inc.”; model: BD-20ACV, Chicago, IL, USA.). This high-frequency discharge method is commonly used as the pre-treatment of polymeric materials before printing and lacquering. Samples were cut into rectangles of 18 × 18 cm and washed with water and ethanol prior to execution. Treatment was conducted in an oscillating voltage output in the range between 10 kV and 45 kV and at 4.5 MHz. The treatment probe length of the electrode was 7 cm, the height between the electrode and sample was approximately 1 cm, and the speed of the manual treatment was around 50 cm/s. Samples were dried in a ventilated climatic chamber (Memmert HPP 110, Schwabach, Germany), set up at 40 °C and 70% RH during 48 h.

#### 2.2.2. Gas Permeance

The gas (O_2_ and CO_2_) permeance (*q*) of all samples was tested at 20 °C, 40 °C, and 60 °C using a manometric method on a Brugger GDP-C appliance [21]. All analyses were performed in triplicate.

#### 2.2.3. Water Vapour Transmission Rate

A modified ASTM E96-80 standard method was used to determine the water vapour transmission rate (WVTR) [22,23]. It was set up with a relative humidity difference of 70%, and the measurement temperature was set to 20 °C. A ventilated climatic chamber was used to store the glass cells (Memmert HPP110, Schwabach, Germany). The slope (*G/t* in g/s) graph and the difference in pressure between both sides of the sample (Δ*p*, Pa) were tracked. The water vapour transmission rate (*WVTR*, g/m^2^·s) was calculated from the change in the cell weight over time at steady state, as follows in Equation (1):(1)$$WVTR=\frac{G}{t\cdot A}$$

#### 2.2.4. The Overall Migration

The total amount of chemical migrants (the overall migration, OM) was evaluated for PET-O/PP laminate (referent commercial sample). Food simulants were chosen according to EU regulations and specific product properties [24]. Thus, 20% (*v*/*v*) ethanol (EtOH) and 3% (*w*/*v*) aqueous acetic acid (HAc) were used. An immersion method (EN 1186-1 Standard) using MigraCell^®^ (FABES Forschungs-GmbH, Munich, Germany) was used during measurements [25]. The migration cells were stored in a controlled condition (Memmert, HPP110, Schwabach, Germany) at 40 °C for 10 days. Subsequently, the food simulants were evaporated at high temperatures (>300 °C) in a previously weighed glass cell (*m*_1_, mg). As soon as all the solution was evaporated, the testing cell was dried to a constant weight at 105 °C (*m*_2_, mg). The *OM* (mg/dm^2^) was based on triplicate measurements and calculated as follows in Equation (2):(2)$$OM=\frac{m_{2}-m_{1}}{A}$$

#### 2.2.5. Fourier Transform Infrared Spectrometry

Fourier transform infrared spectrometry (FTIR, Perkin Elmer FRONTIER spectrum 10, Waltham, MA, USA) was performed to check possible structural changes on the polymer surface after the exposure to food simulants. Both PET-O/PP film sides were tested; however, as the part in direct contact with the food simulant, and later with the food product, is of the main concern, then only results from the PP side are presented. The FTIR spectra were recorded in the frequency range from 4000 to 400 cm^−1^, using ATR (attenuated total reflectance) with a ZnSe crystal. For each measurement, 64 scans with a 4 cm^−1^ resolution were conducted. Spectra were collected in duplicate.

#### 2.2.6. Optical Properties

The opacity was measured spectrophotometrically (UV/VIS Perkin Elmer LAMBDA 25, Waltham, MA, USA), in accordance with [26]. Spectra were recorded in the range from 200 to 800 nm, and the T_600_ value was calculated from the absorbance at 600 nm (A_600_), divided by the sample thickness (in mm). The film colour was measured with a colorimeter (Konica Minolta Spectrophotometer CM3500d, Munich, Germany), using the CIE *L*a*b* colour scale. The following colour parameters were measured: *L** (lightness), *a** (redness), and *b** (yellowness), and the overall difference of colour, Δ*E*, was calculated.

#### 2.2.7. Antimicrobial Activity

*Escherichia coli* ATCC 25922 (Microbial Library, Food Control Center, Zagreb, Croatia) was selected as a non-pathogenic model of undesirable microbial contaminants in the food industry. In Petri dishes containing selective agar, individual colonies of cryopreserved bacteria were isolated and plated. In optimal temperature conditions (38 °C), bacteria were cultured overnight (12 h). The cells were then inoculated with an aliquot of the pre-culture and incubated at 38 °C for 24 h. The disc diffusion method was used for all analyses. Film samples were laid down on the inoculated agar. Film without coating was used as a control sample. Colonies were measured in radial diameter (mm) to estimate relative growth in active film versus the control. The control was performed in an inoculated Petri dish covered with filter paper discs or with pure PET-O/PP. No inhibition zone for control samples was noticed.

### 2.3. Statistical Analysis

Analyses of the data were performed using Xlstat-Pro (win) 7.5.3. (Addinsoft, New York, NY, USA). A one-way analysis of variance (ANOVA) and Tukey’s multiple comparison tests were performed on all data, and the statistical differences were evaluated upon the ranks. Significant results are determined at the *p* > 0.05 confidence level.

## 3. Results and Discussion

In the food packaging industry, the temperature dependency of gas permeability is an important issue. Thus, O_2_ and CO_2_ permeance were measured at different temperatures in referent and Chi-coated PET-O/PP films. The permeance of PET-O/PP (referent) increased with the temperature increase (Figure 1). This was due to the enhanced motion of segments in polymers, as well as to the increased energy level of permeating gas molecules, which makes the diffusion of the penetrant easier. This effect was previously reported in the scientific literature [27,28,29,30]. Furthermore, the orientation of molecules in the polymer matrix could reduce the gas diffusivity, which is the predominant factor for the permeation of gases in polymers [31,32].

In this case, the uniaxially drawn oriented PET layer in the referent commercial sample (PET-O/PP) acts as a barrier because of the film orientation during processing. It was shown by [33] that the oxygen permeability decreased with the increasing draw ratio. Connecting channels between free volume holes were believed to be controlled by the reduced mobility of stretched tie chains.

In the Chi-coated samples, the gas permeance values were significantly decreased (Table 2). For oxygen, the decrease was more than 70%, and for carbon dioxide, it was around 50% for Chi-coated samples and more than 70% for formulations with OPEO. This was attributed to the presence of the functional barrier chitosan coating. The addition of hydrophobic OPEO to the chitosan coating layer led to an even more significant drop in CO_2_ *q* values. Similarly, other authors [34] found that the addition of lemongrass essential oils reduced the carbon dioxide permeability of whey protein-based films. It is generally accepted that the carbon dioxide permeability of biocoatings is dependent on their chemical nature together with the solubility of CO_2_ in the lipid phase, and this versatility is the main reason why the permeation of carbon dioxide may vary. The addition of 1% (*w*/*v*) OPEO to the Chi-coating layer showed a decrease in the permeance values of 67% and 48% for CO_2_ and O_2_, respectively. The decreasing effect seemed not to be proportional to the quantity of the added oil; however, it can be explained as being due to the presence of the mostly impermeable chitosan layer. In addition, it was previously shown in the literature that various essential oils were homogenously distributed across the chitosan film as a result of the flocculation and coalescence during film drying [35]. Further, the presence of essential oils was reported to change the structural organisation of the chitosan and sodium caseinate blend, but without affecting the permeability properties [36]. Then, changes in permeance in the present study were due to the compact chitosan layer, while creating a homogenously distributed essential oil matrix led to a decreasing effect in the final coated material. It is well known that the acetylated units of chitosan can be involved in intra-molecular hydrogen bonding between chitosan chains and/or chitosan and essential oil, creating limited polymer chain motion, which results in relatively low gas permeability [37].

Opposite to gas permeance, the water vapour transmission rate (*WVTR*) in Chi-coated samples was higher compared to PET-O/PP (Table 3). The *WVTR* was around 1.3·10^−5^ (g/m^2^·s), similarly to data previously given in the literature [38].

As can be seen from Table 1, Chi-coated samples contain glycerol in their formulation. Generally, it is well known that the presence of plasticizers increases gas permeability through polymeric materials. The plasticizer composition, molecular weight, and compatibility with the polymer affect the degree of flexibility of the polymer and the permeability of water and oxygen through the plasticized polymer [39,40]. However, it was shown [41,42] that the glycerol (Glyc) can act both as plasticizer or antiplasticizer, depending on its content. An antiplasticizer effect is shown at low glycerol content (≤10%), due to the decreased local dipolar relaxation of the amorphous zein matrix, while at high content (≥20%), glycerol increased the local dipolar relaxation and acts as a plasticizer. The authors speculated that the permeability was regulated mainly by decreasing oxygen solubility rather than decreasing the oxygen diffusion rate since the overall increase in matrix mobility due to glycerol is expected to also increase the oxygen diffusion rate and, thus, increase permeability. Other authors [43] showed that water vapour permeation increased with increasing concentrations of plasticizers (glycerol and PEG), with glycerol increasing the most (from 8.3 to 194.4 g/m s Pa) when the concentration increased from 15% to 45% (*w*/*w*). The author postulated that this is probably due to the increase in free volume in the polymeric network and the decrease in the direct interactions between the chains, which also modify the water vapor diffusion behaviour through the matrix. Consequently, the polymer networks become less dense, promoting the adsorption of water molecules on the surface of the film (higher solubility) and easier penetration through its structure (higher diffusivity), resulting in increased WVP [44,45]. Others [46] also observed that the *WVTR* increased with increasing plasticizer (sorbitol and glycerol) concentration. In contrast, the oxygen transmission rate declined with increasing glycerol concentrations, while no trend was observed for sorbitol. An increased plasticizer concentration from 10 to 15% in the film suspension caused an increase in water vapor permeability from 116% for glycerol-plasticized film and 87.8% for sorbitol-plasticized sweet potato starch film. The WVP generally occurs through the hydrophilic part of the film and, thus, depends on the hydrophilic/hydrophobic ratio of the constituents of the film [47].

Statistically, no significant differences were observed for WVTR among samples. In the literature, various data are given for the water vapour barrier property of films enriched with different active substances, especially in the case of hydrophobic ones, such as the essential oil used in this study [36,48,49,50,51]. Higher permeation values are often attributed to the presence of hydrophilic biopolymer coating [52]. Indeed, permeation is usually greater in the hydrophilic parts compared with the hydrophobic regions. Then, it would not even be surprising that in Chi-coated samples, higher condensation on the tight surface between the supporting polymer layer that is hydrophobic and the chitosan layer that is hydrophilic might lead to the higher mobility of water vapour molecules and, thus, faster permeation, as seen from higher WVTR values (Table 3). In contrast to the given results, others [53] found that PET/PP food-grade films impregnated with olive leaf extract had a lower WVP after the impregnation. Differences from the present study can be explained by the fact that the used substance was hydrophobic, contrarily to chitosan films that are hydrophilic.

Very often it is difficult to compare varying barrier properties of materials in the literature. This could be due to the difference of material compositions, processing histories, aging, test methods, and the accuracy of measurements.

An attempt was made to classify the barrier properties of polymeric packaging materials into five categories [54]. According to this classification, dry ethylene vinyl alcohol, with values of <40, falls into the very high oxygen (cm^3^·μm/m^2^·day·atm) and WVP (g·μm/m^2^·day·kPa) barrier category. Obtained water vapour transfer rate values for the analysed samples were in the range of very high water vapour barriers.

The overall migration values are shown in Figure 2. The simulants 3% acetic acid in water and 20% ethanol in water were considered as food simulants for aqueous acidic (pH ≤ 4.5) and alcoholic foods with an alcohol content of up to 20%, respectively. The OM for 3% HAc and 20% EtOH were 1.8 and 2.3 mg/dm^2^, respectively, which was below the legislation limit of 10 mg/dm^2^ [24]. These OM values in the acid food simulant were lower than given by [55]. OM values on tested commercial multilayer plastics (polystyrene, PS; high impact polystyrene, HIPS; and polypropylene, PP) into 3% aqueous acetic acid simulant were also much lower (from 0.25 to 0.41 mg/dm^2^) than the legislation limit of 10 mg/dm^2^ [56].

Figure 3 shows the FTIR spectra of the PP side of the PET-O/PP film, before and after the exposure to EtOH and HAc. No significant changes in the structural properties of the PP surface were observed, regardless of the simulant used. In all samples, characteristic PP bands were observed [57,58].

The measured colour parameters are given in Table 4. A significant change in all parameters (*L**, *a**, and *b**) could be seen for all Chi-coated samples. After the coating process, changes in the yellowness of the supporting PET-O/PP referent commercial sample occurred. This was attributed to the presence of the chitosan layer that naturally forms yellow films, as well as to the presence of an emulsifier and essential oil also contributing to a yellowish taint (decrease in *a** and *b**). Film pictures are given in Figure 4. Even though such colour difference was visible by the naked eye, from the calculation of the Δ*E* parameter for some samples, the difference was just around a value of 2, but for those enriched with oils, above 3. The total colour difference (Δ*E*) was increased with the addition of higher quantities of emulsifier and essential oil.

No differences in light transmission measured from 200 to 800 nm were noticed (Figure 5). The appearance of small peaks from 200 to 300 nm was attributed to the noise and sensitivity/detectability of the lamp at these small wavelengths. All absorbance values measured at 600 nm (T_600_) were from 1 to 2%, indicating that the films remained transparent even after the Chi-coating process.

The antimicrobial tests showed an inhibition zone of OPEO around 2 mm (Figure 6a). The inhibition was attributed to the antimicrobial activity of the OPEO constituents (D-limonene as a major compound). Samples containing the highest concentration of OPEO (4% (*w*/*v*), sample C/OPEO) did not show any measurable inhibition zone (the figure flash indicates a slight inhibition zone that was not continuous, thus it was not taken into account) (Figure 6b). White circles under the sample (Figure 6b) cannot be attributed to the inhibitory action of the sample, but to the fact that the sample was directly put on the surface. Even though that *E. coli* is a facultative anaerobe, it can metabolise more energy from its nutrient source and grow faster if oxygen is present [59], so the authors suppose that in those specific regions, lack of oxygen was the reason for *E. coli* not growing under the sample. The lack of inhibition was attributed to the insufficient amount of active compound (OPEO) needed to achieve the bacteriostatic effect.

## 4. Conclusions

A fruit-based active compound, the essential oil extracted from orange waste, was used to enrich the chitosan-based coating on a commercial polymeric film. The gas permeance increased with increasing test temperatures from 20 to 60 °C, for both oxygen and carbon dioxide. The gas permeance of samples with biolayers was significantly decreased. Adding a thin chitosan layer (20–35% mass proportion to PET-O/PP) did not significantly change the water vapour barrier (given as WVTR), so as such, it might be used to improve the gas barrier of commercial film without impacting its water vapour barrier. The base layer of PET-O/PP (referent commercial sample) was exposed to various food simulants, and based on FTIR analyses, no significant changes of the PP surface occurred. Samples were also confirmed to be adequate for food use since the overall migration measured was below the legislative migration limits. Regarding the transparency, no noticeable changes were observed at 600 nm for Chi-coated samples in the range of 1 to 2% OLEO. Finally, antimicrobial tests performed with formulations enriched with 4% OPEO showed no activity against *E. coli*, indicating that the added amount of oil was not sufficient for obtaining the desired bacteriostatic effect, and therefore, further research with higher concentrations and/or other formulations is required. Use of a Chi-coating enriched with OPEO made from natural waste can potentially offer improved gas barrier properties without affecting optical and water vapour properties. Thus, it might be considered as a potential path in a green approach towards a sustainable packaging.

## Figures and Tables

**Figure 1 polymers-15-02569-f001:**
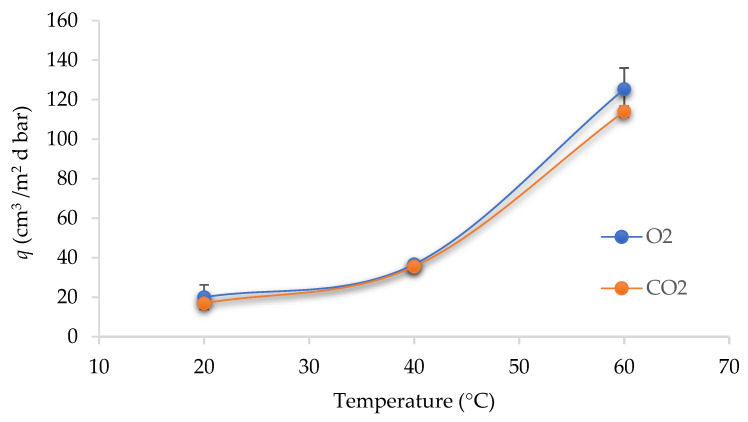
Permeance (*q*) values through PET-O/PP (12/50 µm) referent commercial sample at different temperatures.

**Figure 2 polymers-15-02569-f002:**
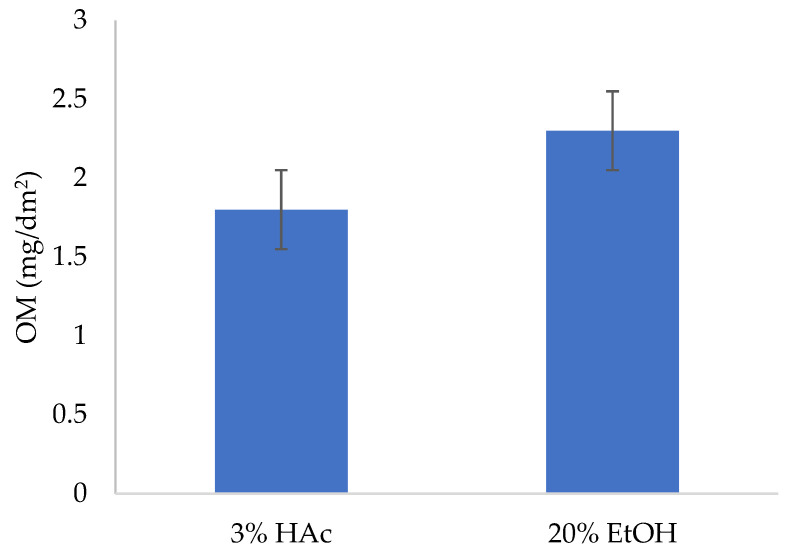
The overall migration values from PET-O/PP referent commercial sample into food simulants: 3% (*w*/*v*) acetic acid (HAc) and 20% *v*/*v* aqueous ethanol (EtOH).

**Figure 3 polymers-15-02569-f003:**
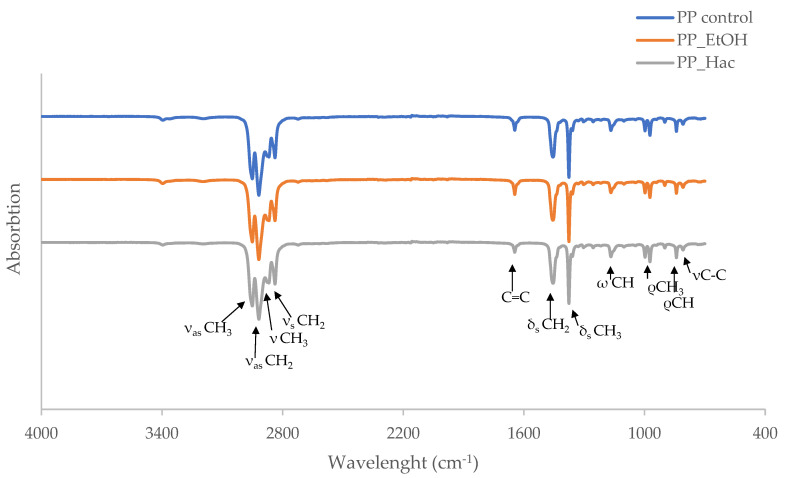
FTIR spectra of PET-O/PP (referent commercial sample; facing the PP side of the laminate) before and after exposure to food simulants.

**Figure 4 polymers-15-02569-f004:**
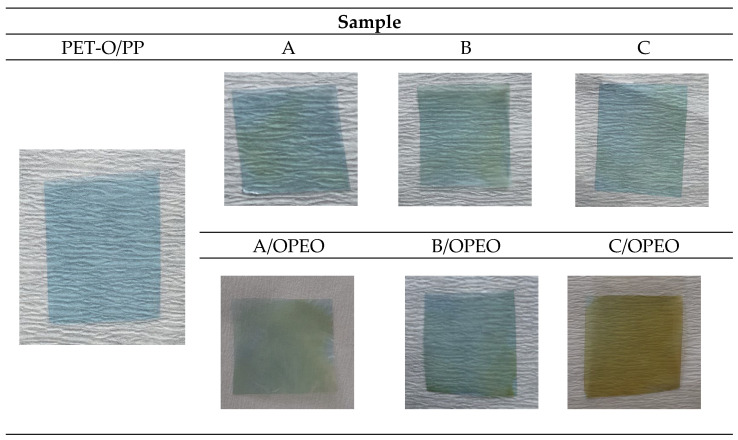
Pictures of films taken on the white surface.

**Figure 5 polymers-15-02569-f005:**
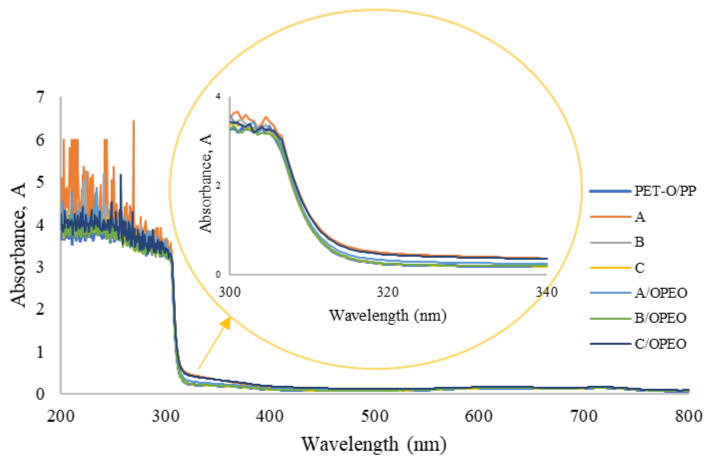
Absorbance (A) values for plain PET-O/PP referent commercial sample and the Chi-coatings enriched with orange peel essential oil (OPEO): 1% *w*/*v* (A), 2% *w*/*v* (B), and 4% *w*/*v* (C).

**Figure 6 polymers-15-02569-f006:**
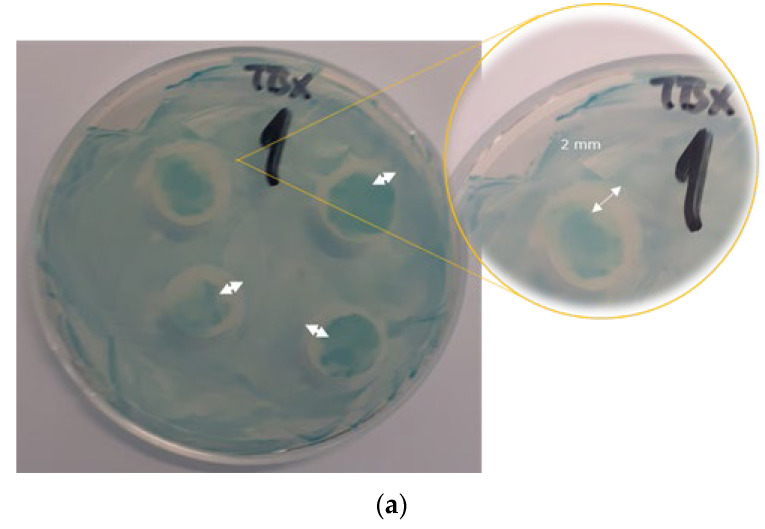
Effect of the PET-O/PP film coated with chitosan enriched with 4% OPEO on the growth of *Escherichia coli* (ATTC 25922). (**a**) Orange peel essential oil (OPEO), 100 µL, and (**b**) PET-O/PP referent commercial sample with Chi-coating enriched with 4% OPEO (C/OPEO).

**Table 1 polymers-15-02569-t001:** Formulation of chitosan (Chi) coatings.

Sample Composition	Chitosan (%, *w*/*v*)	Tween 80 (%, *w*/*v*)	Glycerol (w/pdw)	OPEO (*w*/*v*)	Coating layer Thickness (µm)	Coating Layer Proportion (Mass)
A	2	0.3	30	-	19	20%
B	2	0.6	30	-	25	20%
C	2	1.2	30	-	32	20%
A/OPEO	2	0.3	30	1%	20	25%
B/OPEO	2	0.6	30	2%	22	25%
C/OPEO	2	1.2	30	4%	37	30%

OPEO—Orange peel essential oil; w/pdw—weight/polymer dry weight.

**Table 2 polymers-15-02569-t002:** Gas permeance (*q*) of plain and Chi-coated PET-O/PP referent commercial sample.

Sample Composition	*q* (cm^3^/m^2^·d·bar)	
	O_2_	CO_2_
PET-O/PP	20.03 ± 6.20 ^a^	16.90 ± 1.06 ^a^
A	4.53 ± 2.82 ^b^	8.06 ± 0.00 ^b^
C	6.00 ± 2.12 ^b^	7.74 ± 0.07 ^b^
A/OPEO	2.25 ± 0.70 ^b^	2.83 ± 0.14 ^c^
C/OPEO	3.38 ± 1.41 ^b^	5.07 ± 0.21 ^c^

OPEO-Orange peel essential oil; Different superscripts (a–c) within the same column indicate significant differences among samples (*p* < 0.05).

**Table 3 polymers-15-02569-t003:** Water vapour transmission rate (*WVTR*) for referent commercial sample and Chi-coated PET-O/PP.

Samples	*WVTR* ·10^−5^ (g/m^2^·s)
PET-O/PP	1.27 ± 0.45
A	5.89 ± 0.60
B	3.31 ± 0.34
C	9.50 ± 6.33
A/OPEO	3.58 ± 0.58
B/OPEO	4.09 ± 0.39
C/OPEO	5.32 ± 0.29

OPEO—Orange peel essential oil.

**Table 4 polymers-15-02569-t004:** CIE *L*a*b* colour parameters and light transmission for referent commercial sample PET-O/PP and Chi-coated (A, B, C) PET-O/PP film.

Samples	*L**	*a**	*b**	Δ*E*	T_600_
PET-O/PP	87.75 ± 0.10 ^a^	−3.68 ± 0.10 ^a^	−14.47 ± 0.16 ^c^	0.0±0.0 ^d^	1.97
A	87.07 ± 0.21 ^b,c^	−4.60 ± 0.36 ^b,c^	−12.40 ± 0.77 ^b^	2.4± 0.9 ^a^	1.61
B	87.25 ± 0.13 ^a,b^	−4.39 ± 0.19 ^b^	−12.62 ± 0.42 ^b^	2.0± 0.5 ^a^	1.41
C	86.51 ± 0.75 ^d,e^	−5.02 ± 0.46 ^b,c^	−11.53 ± 1.00 ^a,b^	3.5± 1.2 ^b,c^	1.31
A/OPEO	86.92 ± 0.05 ^b,c,d^	−4.83 ± 0.28 ^b,c^	−11.43 ± 0.38 ^a,b^	3.4± 0.4 ^a,b^	1.63
B/OPEO	86.64 ± 0.30 ^c,d^	−4.81 ± 0.42 ^b,c^	−12.44 ± 1.15 ^b^	2.6±1.2 ^a,b^	1.56
C/OPEO	85.98 ± 0.31 ^e^	−5.90 ± 0.39 ^d^	−9.90 ± 1.33 ^a^	5.2±1.0 ^c^	1.80

OPEO—Orange peel essential oil; Different superscripts (a–e) within the same column indicate significant differences among samples (*p* < 0.05).

## Data Availability

Data supporting the reported results can be found at the authors’ repository.

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
