# Peer review of "Characterization of Synthetic Polymer Coated with Biopolymer Layer with Natural Orange Peel Extract Aimed for Food Packaging"

_polymers, 2023, doi:10.3390/polym15112569_

Round 1
Reviewer 1 Report (Previous Reviewer 1)
In general, the quality of the manuscript was improved; however, it still is not in the version to be accepted.
The title changed to “Characterization of synthetic polymer coated with biopolymer layer with natural orange peel compounds aimed for food packaging.” However, the research does not report the identification of orange peel compounds. In the results section, it is only once mentioned that D-limonene is the major compound, with no information about its antimicrobial activity.
Introduction: There is no background on the antimicrobial activity of OPEO. It is well known at the scientific level that chitosan also presents antimicrobial activity, but it is not commented on in the manuscript.
The description of the aims of the research was improved.
The materials description was corrected as requested for the thickness of the commercial bilayer film.
Comments to coatings composition section. The authors presented the final composition of the coating formulations in Table 1. They completed the description by adding the coating thicknesses and their proportions in the films. But still is not clear the final concentration of the OPEO in the coatings of the different treatments. Some components are expressed in w/v and plasticizer in w/pdw. It is also indicated that initially, a solution was prepared, applied on the PP layer, and then the liquid was evaporated for 48 h. However, they did not inform the estimated final concentration of the OPEO in the dry coating of each treatment once the solvent was evaporated. It seems to be much more than the 1-4% mentioned for the liquid formulations. This needs to be clarified. What about if the components of the OPEO were evaporated during the 48 h drying?
Comments on the permeability, permeance, and transmission rate of gases: the presentation and discussion of results were improved. The authors eliminated part of the permeability and activation energy (Ea) sections from the first version of the manuscript. Although the property permeability is still mentioned for the multilaminate film (see line 225), it should change to permeance.
Lines 228-229. Between which molecules the mentioned hydrogen bonds are formed?
Comments on the permeability, permeance, and transmission rate of water vapor: The way to express the transference of water vapor through a multilayer film in Table 3 is permeance or transmission rate. Thus, only the second column in this table is correct. Although the authors eliminated part of the permeability and activation energy (Ea) sections from the first version of the manuscript, they still do not understand that it is not possible to calculate water vapor permeability for a bilayer or multilayer film. Each component of the structure has its own permeability and cannot be integrated because the thickness of PP cannot be just added to that of the PET layer. This also applies to the coating layer.
Lines 245-295. The discussion and explanation of the effect of the coating components on the permeance or permeability is confusing. The authors described how other works explained the effect, but a very short explanation is given for their own materials. Moreover, the authors are still presenting water vapor permeability for a multilayer film. As explained before, this is not possible; how did they incorporate the thickness value in the permeability values?
Comments on the overall migration section: Authors attended the suggestion and changed the references for those that reported migration in the same unities. However, which was the justification to evaluate overall migration in the commercial film, it is supposed that if it is in the market is because it complies with the overall migration limit established in the food contact regulation (this comment also applies to the conclusions section, lines 374-375). Why the coated film was not tested for overall migration? That would have been interesting to know how much the chitosan coating increased the migration. Additionally, because of the properties of PP, this layer is not expected to be affected by acid or ethanolic solutions, as shown by the FTIR analysis. The chitosan-glycerol-emulsifier layer could be solubilized in both simulants. Chitosan is soluble in acetic acid solutions and glycerol in ethanolic solutions. Those results need to be included in the research.
Comments on the effect of the coatings on the films’ color and UV VIS absorbance. The authors found no effect on the coatings from 200-800 nm. However, this effect is better observed in the infrared section of the electromagnetic spectra; thus, the FTIR analysis could have shown chemical groups of essential oil components, like double bonds in limonene. Additionally, FTIR can show the OH in glycerol, NH2 in chitosan, etc.
Antimicrobial activity of OPEO. The authors did not make a film with a higher OPEO concentration coating, as recommended. In the methodology section (lines 180-182), the authors described the technique as “To estimate the relative growth of cells in presence of active film compared to the control, the radial diameter (mm) of colonies was measured.” According to the results, they did not measure the diameter of colonies but the area where there was no growth of E. coli. This paragraph needs revision and correction.
Fig. 6a showed a 2 mm inhibition zone around the OPEO, but the authors did not include a control sample (or blank) on the petri dish, to compare the inhibition zone size. At least is not mentioned in the methodology.
As Fig. 6 does not show antimicrobial activity, the caption should change to "Effect of the PET-O/PP film coated with chitosan enriched with 4% OPEO on the growth of Escherichia coli (ATTC 25922)."
Author Response
Dear Editor,
Thank you for your and reviewer’s’ comments.
Authors followed all the recommendations suggested by reviewers that are detailed down below. In the Revised Manuscript file, all the changes are highlighted in yellow. In addition, changes are made following repetition rate report for the sections where it was noted as too much of overlapping. Since new references are added, and one deleted, the final number of references is changed to 59.
Reviewers comments
Reviewer 1
Comments and Suggestions for Authors
In general, the quality of the manuscript was improved; however, it still is not in the version to be accepted.
Comment
The title changed to “Characterization of synthetic polymer coated with biopolymer layer with natural orange peel compounds aimed for food packaging.” However, the research does not report the identification of orange peel compounds. In the results section, it is only once mentioned that D-limonene is the major compound, with no information about its antimicrobial activity.
Answer
Authors thank for the comment. Unfortunately, we were not able to perform orange peel compounds identification. Orange peel was extracted using MAE and the scope was to investigate only the possibility of applying the OPEO and not going in deeper investigation of D-limonene.
Comment
Introduction: There is no background on the antimicrobial activity of OPEO. It is well known at the scientific level that chitosan also presents antimicrobial activity, but it is not commented on in the manuscript.
Answer
Following information on chitosan and OPEO antimicrobial activity is added.
Antimicrobial activity of chitosan films was observed on the growth of a wide variety of bacteria (Bacillus cereus, Escherichia coli, Salmonella typhimurium, Staphylococcus aureus, Listeria monocytogenes etc.) and fungi (Botrytis cinerea, Fusarium oxysporum, Trichophyton equinum, Piricularia oryzae etc.) [16,17].
Phenols in citrus peels are known to have antimicrobial activity due to their high amounts of polyphenols and were found to be effective against Staphylococcus aureus, Escherichia coli, Pseudomonas aeruginosa and Klebsiella pneumoniae [18] as well as against food-borne pathogens [19].
Thus, new references are added:
- Rodríguez-Núñez, J.R.; Madera-Santana, T.J.; Sánchez-Machado, D.I.; López-Cervantes, J.; Soto Valdez, H. Chitosan/hydrophilic plasticizer-based films: preparation, physicochemical and antimicrobial properties. Journal of Polymers and the Environment 2013, 22, 41-51, doi:10.1007/s10924-013-0621-z.
- Goy, R.C.; Britto, D.d.; Assis, O.B.G. A review of the antimicrobial activity of chitosan. Polímeros 2009, 19, 241-247, doi:10.1590/s0104-14282009000300013.
- Yashaswini, P.; vind, A. Antimicrobial properties of orange (Citrus reticulata var. Kinnow) peel extracts against pathogenic bacteria. International Journal of Current Microbiology and Applied Sciences 2018, 7, 737-746, doi:10.20546/ijcmas.2018.703.086.
- Shehata, M.G.; Awad, T.S.; Asker, D.; El Sohaimy, S.A.; Abd El- Aziz, N.M.; Youssef, M.M. Antioxidant and antimicrobial activities and UPLC-ESI-MS/MS polyphenolic profile of sweet orange peel extracts. Current Research in Food Science 2021, 4, 326-335, doi:10.1016/j.crfs.2021.05.001.
Comment
The description of the aims of the research was improved.
Answer
Authors thank for the affirmative comment.
Comment
The materials description was corrected as requested for the thickness of the commercial bilayer film.
Answer
Authors thank for the affirmative comment.
Comment
Comments to coatings composition section. The authors presented the final composition of the coating formulations in Table 1. They completed the description by adding the coating thicknesses and their proportions in the films. But still is not clear the final concentration of the OPEO in the coatings of the different treatments. Some components are expressed in w/v and plasticizer in w/pdw. It is also indicated that initially, a solution was prepared, applied on the PP layer, and then the liquid was evaporated for 48 h. However, they did not inform the estimated final concentration of the OPEO in the dry coating of each treatment once the solvent was evaporated. It seems to be much more than the 1-4% mentioned for the liquid formulations. This needs to be clarified. What about if the components of the OPEO were evaporated during the 48 h drying?
Answer
Authors understand doubts about the concentration after drying. Indeed, and generally, in these kinds of systems there is a certain loss of volatiles during drying. Even though the biopolymer is aimed to protect the volatile by its encapsulation and bonding, in most cases and according to literature the protective role is not efficient 100%. Also, as we have not possibility to perform the determination of OPEO concentration, we named samples by the initial/added composition. This is also often done as such in the literature, and no many research studies provide the concentration by the end of drying.
Comment
Comments on the permeability, permeance, and transmission rate of gases: the presentation and discussion of results were improved. The authors eliminated part of the permeability and activation energy (Ea) sections from the first version of the manuscript. Although the property permeability is still mentioned for the multilaminate film (see line 225), it should change to permeance.
Answer
Thanks for your observation. The mentioned omission is corrected.
Comment
Lines 228-229. Between which molecules the mentioned hydrogen bonds are formed?
Answer
The explanation is provided and changed in text as follows:
It is well known that the acetylated units of chitosan can be involved in intra-molecular hydrogen bonding between chitosan chains and/or chitosan and essential oil creating limited polymer chain motion, which results in relatively low gas permeability [37].
Comment
Comments on the permeability, permeance, and transmission rate of water vapor: The way to express the transference of water vapor through a multilayer film in Table 3 is permeance or transmission rate. Thus, only the second column in this table is correct. Although the authors eliminated part of the permeability and activation energy (Ea) sections from the first version of the manuscript, they still do not understand that it is not possible to calculate water vapor permeability for a bilayer or multilayer film. Each component of the structure has its own permeability and cannot be integrated because the thickness of PP cannot be just added to that of the PET layer. This also applies to the coating layer.
Lines 245-295. The discussion and explanation of the effect of the coating components on the permeance or permeability is confusing. The authors described how other works explained the effect, but a very short explanation is given for their own materials. Moreover, the authors are still presenting water vapor permeability for a multilayer film. As explained before, this is not possible; how did they incorporate the thickness value in the permeability values?
Answer
Thanks for your comments. Our intention was to present both WVP and WVTR values just to be easily compared with research from other authors. However, we agree with your observation and only WVTR value is now presented in Table 3.
As for the lines 245-295, an inserted discussion is done with regards to previous reviewer comment from 1st revision („It seems that the chitosan material presented an important effect on the oxygen and carbon dioxide permeability. It is recommended to include published data of gases permeability of chitosan. How the glycerol additive could affect the diffusivity of these two gases?“)
Comment
Comments on the overall migration section: Authors attended the suggestion and changed the references for those that reported migration in the same unities. However, which was the justification to evaluate overall migration in the commercial film, it is supposed that if it is in the market is because it complies with the overall migration limit established in the food contact regulation (this comment also applies to the conclusions section, lines 374-375). Why the coated film was not tested for overall migration? That would have been interesting to know how much the chitosan coating increased the migration. Additionally, because of the properties of PP, this layer is not expected to be affected by acid or ethanolic solutions, as shown by the FTIR analysis. The chitosan-glycerol-emulsifier layer could be solubilized in both simulants. Chitosan is soluble in acetic acid solutions and glycerol in ethanolic solutions. Those results need to be included in the research.
Answer
We agree with you that all packaging materials placed on the market should comply with legislation and be safe in contact with food. However, practically this is usually done as there were cases when the values were above legislation limit.
Coated films were not analyzed as all components are edible and thus of no safety concern. Following comment was sent to reviewer in comments of 1st revision: “OPEO is a natural product (all components are edible). OPEO will be dissolved in food simulants thus giving unrealistic situation. Unlike natural product the concern in this study was on the synthetic material.”
Comment
Comments on the effect of the coatings on the films’ color and UV VIS absorbance. The authors found no effect on the coatings from 200-800 nm. However, this effect is better observed in the infrared section of the electromagnetic spectra; thus, the FTIR analysis could have shown chemical groups of essential oil components, like double bonds in limonene. Additionally, FTIR can show the OH in glycerol, NH2 in chitosan, etc.
Antimicrobial activity of OPEO. The authors did not make a film with a higher OPEO concentration coating, as recommended. In the methodology section (lines 180-182), the authors described the technique as “To estimate the relative growth of cells in presence of active film compared to the control, the radial diameter (mm) of colonies was measured.” According to the results, they did not measure the diameter of colonies but the area where there was no growth of E. coli. This paragraph needs revision and correction.
Answer
Comment to reviewers after 1st revision: ”Authors thank for this suggestion as this was also what we wanted to do. As we faced some technical problem, we were not able to continue with this plan. However, our intention is to do it together with additional formulations as soon as situation allowed us.”
Comment
Fig. 6a showed a 2 mm inhibition zone around the OPEO, but the authors did not include a control sample (or blank) on the petri dish, to compare the inhibition zone size. At least is not mentioned in the methodology.
Answer
The control sample/blank did not show any inhibition zone. Explanation is added to the materials and methods section as:
Lines 184-187
The control was performed in inoculated Petri covered with filter paper discs or with pure PET-O/PP.
Comment
As Fig. 6 does not show antimicrobial activity, the caption should change to "Effect of the PET-O/PP film coated with chitosan enriched with 4% OPEO on the growth of Escherichia coli (ATTC 25922)."
Answer
Authors thank for this comment. Indeed, it is more appropriate to write figure caption as:
Figure 6. Effect of the PET-O/PP film coated with chitosan enriched with 4% OPEO on the growth of Escherichia coli (ATTC 25922).

Reviewer 2 Report (Previous Reviewer 2)
Fine work
Author Response
Authors are thankful for reviewer’s acceptation of modifications.

Reviewer 3 Report (Previous Reviewer 3)
I appreciate the great work done by the authors.
English is satisfactorily
Author Response
Authors are thankful for reviewer’s acceptation of modifications.

Round 2
Reviewer 1 Report (Previous Reviewer 1)
The manuscript has improved. Authors responded most of the suggestions or justified why they did not accept them. Regarding to those that were not incorporated in the manuscript, I have the following recommendations. I am sure that if they are considered, the quality will increase.
If authors were not able to perform orange peel compounds identification, it is recommended to change the title to "Characterization of synthetic polymer coated with biopolymer layer with natural orange peel extract (or oil, or essential oil) aimed for food packaging".
Table 1. There is no intention to present the composition of the coating, even as an estimation after evaporating the solvent. Solvent that was never informed wish was.
Line 131. subtitle should be "Water vapour transmission rate".
Table 3. It was modified as recommended. But now I noticed that the standard deviation is huge, much higher than the averages in same cases. See an average of 9.5 and standard deviation of 66.33 for sample C. Check the results and if they are correct, is better to use ranks instead of averages and standard deviation.
Author Response
Reviewer 1
The manuscript has improved. Authors responded most of the suggestions or justified why they did not accept them. Regarding to those that were not incorporated in the manuscript, I have the following recommendations. I am sure that if they are considered, the quality will increase.
Authors are thankful for positive feedback. We appreciate reviewer’s precious time taken to improve the quality of the manuscript.
Comment
If authors were not able to perform orange peel compounds identification, it is recommended to change the title to "Characterization of synthetic polymer coated with biopolymer layer with natural orange peel extract aimed for food packaging".
Answer
Authors agree with proposed change of the title. It is thus changed to:
"Characterization of synthetic polymer coated with biopolymer layer with natural orange peel extract aimed for food packaging"
Comment
Table 1. There is no intention to present the composition of the coating, even as an estimation after evaporating the solvent. Solvent that was never informed wish was.
Answer
The solvent type is added along with the description of the preparation of coatings.
Biopolymer solutions were prepared from chitosan (at 2% w/v, in 1% aqueous acetic acid (v/v)), glycerol as plasticizer (30% of polymer dry weight (w/pdw)) and Tween 80 as emulsifier.
Comment
Line 131. subtitle should be "Water vapour transmission rate".
Answer
Subtitle is changed to
2.2.3. Water vapour transmission rate
Comment
Table 3. It was modified as recommended. But now I noticed that the standard deviation is huge, much higher than the averages in same cases. See an average of 9.5 and standard deviation of 66.33 for sample C. Check the results and if they are correct, is better to use ranks instead of averages and standard deviation.
Answer
Authors are very thankful for this comment. Actually, there was an error while copying results to word file, so actual values of standard deviation are corrected.
Table 3. Water transmission rate (WVTR) for referent commercial sample and Chi-coated PET-O/PP
|
Samples |
WVTR·10-5 (g/m2·s) |
|
|
PET-O/PP |
1.27 ± 0.45 |
|
|
A |
5.89 ± 0.60 |
|
|
B |
3.31 ± 0.34 |
|
|
C |
9.50 ± 6.33 |
|
|
A/OPEO |
3.58 ± 0.58 |
|
|
B/OPEO |
4.09 ± 0.39 |
|
|
C/OPEO |
5.32 ± 0.29 |
|

Reviewer 3 Report (Previous Reviewer 3)
I appreciate the work of the authors and the results obtained.
Author Response
Comment
I appreciate the work of the authors and the results obtained.
Answer
Authors thanks for the affirmation comment.

This manuscript is a resubmission of an earlier submission. The following is a list of the peer review reports and author responses from that submission.
Round 1
Reviewer 1 Report
The manuscript's name is "Synthetic polymer characteristics improvements with active biopolymer coating aimed for food packaging". The novelty of the research is the implementation of a coating based on chitosan and orange peel essential oil (OPEO) on a commercial lamination: oriented poly(ethylene-terephthalate)/polypropylene (PET-O/PP). The purpose was to develop an antimicrobial film, but at the end there was no antimicrobial effect of the development on the growth of E. coli bacteria. In the introduction, authors also enhanced the importance to develop multilayer materials with at least one layer of biopolymer (or biodegradable composites). However, in the results they did not provide information about the proportion or percentage of the coating as the bio-layer.
There are several observations and recommendations before the acceptation of the manuscript:
Materials and methods:
Line 84. Include the thickness of each polymer layer of the PET-O/PP laminate.
Line 97. The preparation of the coating is not clear: which was the solvent used in the blend? How the formulation was applied on the PP layer of the film? If it was manual, how do they control the thickness? Which was the real concentration of the OPEO on the dry coating of each treatment? Once the solvent was evaporated.
Table 1. Meaning of w/pdm.
Line 120. Check the diffusion coefficient units (cm2·s)
Line 135. If the oxygen and carbon dioxide permeability was measured at 20, 40, and 60 °C (because 20°C was a typical storage temperature, and 40 °C and 60 °C as elevated and inadequate storage temperatures), why the WVTR was tested at 25°C? Why the difference?
Line 147. The methodology did not mention the food product to simulate for overall migration tests. Why the overall migration was not tested in the films coated with the OPEO formulations?
Results
Line 199. It is not usual to start the results section with figures and tables. They must be incorporated where those results are mentioned/discussed. Reorganize. At least that the journal allows such style.
Line 201. Permeance values depend on thickness. Include the thickness of each layer in the figure 1 caption.
Discussion
Lines 259-269. Permeability values are discussed for the bilayer PET-O/PP lamination. However, permeability is the property of one polymer. The unities or permeability include the thickness of the film or laminate. Which thickness the authors used to calculate permeability? If they used the 65 micrometers of the two layers it is not correct. Notice that all values mentioned in the discussion, reported by others, are for monolayer materials. In bilayer or multilayer cases is better to use permeance (q).
Lines 270-294. If the permeability value is not correctly calculated, the Ea is not correct. Moreover, the discussion of the Ea is very basic and not related with the function of the films as food packaging. I mean, there is no interpretation of the Ea results. Which values of Ea are suitable to the rank of temperatures that a food is subjected during packaging, transportation, commercialization and home use? Are these finding appropriate to maintain the modified atmosphere vacuum or air content in the package?
Line 210. Figure 2c. Something is wrong with the CO2 line, only two points are visible and the line is not aligned with them. A third point is missing.
Lines 293-300. Any statistical analysis to confirm the decrease of Permeability values? Table 3 does not show the values for CO2. Table 3 does not show the temperature of measurements.
Lines 300-304. It seems that the chitosan material presented an important effect on the oxygen and carbon dioxide permeability. It is recommended to include published data of gases permeability of chitosan. How the glycerol additive could affect the diffusivity of these two gases?
Line 306. Superscripts do not show significant differences. Or there are mistakes on Table 4 data, or the values are similar.
Line 334. Be careful when comparing OM values. The unities reported by 43 are not mentioned and the ones reported by 44 are not equivalent (mg/kg) to those used in the research (mg/dm2).
Lines 337-345. If there are no changes in the FTIR spectra, there is no need to interpret the bands. It would have been interesting to see the effect of chitosan/Glycerol/Tween and chitosan/Glycerol/Tween with OPEO.
Lines 354-355. According to Table 5, there is no effect of the addition of OPEO on the a*, b* or delta E values, except for the a* value in C/OPEO films.
Line 362. This result was disappointing. Why to write an article if the main objective is not positive? Moreover, chitosan also has antimicrobial activity. It would have been interesting to see a discussion related to a separate or synergistic effect of chitosan and OPEO.
Conclusions
At the end of the conclusions, authors recommended to increase the concentration of the OPEO in the coating to improve the bacteriostatic effect. Therefore, my recommendation is to perform this new stage of the research with higher concentration of OPEO, add it to the manuscript, attend all additional recommendations, and submit it again.
Additional recommendation. Why not to test the antimicrobial packaging with deteriorative bacteria instead of pathogenic? Or with fungus that easily grow in food products.
Reviewer 2 Report
This manuscript reports that a synthetic laminate, oriented poly(ethylene-terephthalate)/polypropylene (PET-O/PP) was coated with orange peel essential oil (OPEO) to enrich the biopolymeric film. The performance of permeation (O2, CO2, light and Water) was also tested. Additionally, the antimicrobial activity was also measured here. Some intereting results were provided, but the shortcomings of this article are also obviously.
1. The contents of this work can not cover this big title, which would mislead the readship.
2. So much contents are present in abstract, which should have only included some interesting points to catching the attention from readers.
3. The background about the serious environmental issues are exactly improtant. But your work’s importance is not clear. And what is the key drawback you find from the recent literature about the “Designing multilayer materials”? Please emphasize the novelty of your research.
4. Discussion section. The phenomenons are not explained very well. For one example, Line 302-304: The decreasing effect seemed not to be proportional to the quantity of the added oil, however it can be explained as due to the presence of oil and possible changes in chitosan/OLEO structure. What kind of structure? In many cases, the structure can affect the preperty of film, but the limited information provided from this article probably challenges the deep understanding of readers. Also line 350-351, Line 364-366.
Reviewer 3 Report
please see the attached file

Reviewer 4 Report
Line 9: This sentence should be rewritten. The polymeric film was coated by biolayer (coating) enriched with essential oil.
Lines 10-14: You add an active bio-based layer to the polymer, however why did you not take in account that this layer is inseparable or hard to separation during the recycling process? The EU regulations, for example Zero Waste 2030 are about sustainable, recyclable materials and how do you plan to recycle this one? The green polymers need special environmental conditions for composting process and simultaneously polyolefins are recovered using mostly the mechanical methods.
The paper doesn't have any practical application. In the next sentences you give overal examples of type of foods and the barrier for gases needed for food protection. Do those food items need a special barrier for selected gases? Is it necessary to add any extra barrier layer? The biocotings have mostly poorer barrier for water vapour, so in my opinion the barrier isn't necessary. Moreover according to UE policy the level of barrier should be optimal - there is no reason behind for adding extra barrier if it is not mandatory.
Line 16: FTIR is not for measuring the surface. This is for material inventory testing.
Keywords: I suggest to be more precise, instead using overall keywords as you have already used.
Line 54: What do you mean by safer?
Line 69: Isn't it water vapour a gas as well?
Line 96: What means OPEO?
Line 160: Could you provide more details here? Which side did you measure? If both then were are the results of both side? If one then were are the results of all samples?
Table 5. Delta E is significant - could you please present the photographs of obtained films?
Diffusion coefficient: could you explain please what means obtained values of DC? The levels of fitting to the model?
Line 296: Could you give the values please for those significant values? Or percentage?
Discussion: I don't see this part here. You describe mostly your results but here is no conclusions, comparisons to other authors and firstly the comments from your side.
The 5th part: conclusions sounds like an abstract. Please rewrite this part.
Some of the references should be replaced, especially those that are more than 10 years.